# Rheological Properties, Particle Size Distribution and Physical Stability of Novel Refined Pumpkin Seed Oil Creams with Oleogel and Lucuma Powder

**DOI:** 10.3390/foods11131844

**Published:** 2022-06-22

**Authors:** Angela Borriello, Nicoletta Antonella Miele, Paolo Masi, Silvana Cavella

**Affiliations:** 1Department of Agricultural Sciences, University of Naples Federico II, 80055 Portici, Italy; angela.borriello@unina.it (A.B.); pmasi@unina.it (P.M.); cavella@unina.it (S.C.); 2Centre of Food Innovation and Development in the Food Industry (CAISIAL), University of Naples Federico II, 80055 Portici, Italy

**Keywords:** healthy creams, oleogels, pumpkin seed oil, carnauba wax, ball mill refining, natural sweetener

## Abstract

This research aimed to develop new hazelnut and pumpkin seed oil-based creams and to assess the effect of different fat and sugar phases on the structure and physical properties of those creams at different refining degrees. In this study, three novel spreadable creams were prepared in a stirred ball-mill: CBS with cocoa butter, pumpkin seed oil and saccharose; OS with pumpkin seed oil and carnauba wax-basedoleogel and saccharose; OLS with oleogel, saccharose and Lucuma powder. OS and CBS creams reached a D90 value lower than 30 µm at 150 min of refining, the OLS cream showed the highest D90 value, with a particle size distribution and a rheological behaviour little affected by the refining time. The OS and CBS creams differed in yield stress, indicating that the attractive particle–particle interactions are affected not only by the particle size, but also by fat composition. Moreover, all the creams showed solid-like behaviour and a good tolerance to deformation rate, a high oil-binding capacity and a good physical stability. Thus, it is possible to reformulate spreadable creams with healthier nutritional profiles.

## 1. Introduction

Sweet anhydrous creams represent an important ingredient in different confectionery foods, but they have often achieved the greatest success on their own as spreads, whose market is constantly developing. Creams are considered unhealthy products due to the large amounts of fat and sugar, which strictly affect their physical and sensory properties. Hard fats commonly used in cream production are rich in saturated fatty acids, whose consumption also leads to negative health implications [1]. It is not suitable to directly replace hard fats with liquid oil, leading to technological problems, such as texture weakness and oil leakage [2]. To solve this problem, in the last few years, the oleogelation process has been successfully developed. The oleogelation process consists of structuring liquid oil by using an oleogelator, offering the functionality of hard fats with the nutritional profile of liquid oil to food products, avoiding saturated and trans-fats. Our group has recently developed new oleogels with pumpkin seed oil and carnauba wax for potential food applications [3]. Pumpkin seed oil was distinguished from other vegetable oils, such as hempseed and almond oils, by the presence of arachidonic acid (0.27%), which is essential for optimal performance of the nervous system [4,5]. On the other hand, the caloric value of a spreadable cream is largely affected by the sugar, generally saccharose, which is a key ingredient for the structure and quality of the product. Daily sugar intake should be less than 10% of total caloric intake, as stated by the WHO guidelines, because its consumption negatively affects health, increasing the risk of several chronic diseases [6]. Several strategies can be adapted to reduce sugar in foodstuffs. The most commonly suggested one is product reformulation by partially/totally replacing sugar, or reducing its amount [7]. For sugar substituting, natural plant-based sweeteners are usually preferred to artificial sweeteners, as they may contain beneficial bioactive compounds, such as polyphenolic compounds with antioxidant properties. Lucuma is a Peruvian fruit with a similar sweet caramel taste that should be used as a natural food sweetener, with a high nutritional value [8,9,10]. It is a low glycaemic sweetener and a useful source of fibre, iron, niacin and beta-carotene, phenols and flavonoid components, thus it could be considered a superfood [11]. Lucuma was successfully introduced in many foodstuffs such as ice cream, juices, cakes, biscuits, yogurt, chocolate, baby food and pies [12,13]. Spreadable cream production consists of mixing and refining the ingredients, obtaining the characteristic structure of a concentrated suspension of both hydrophobic and hydrophilic solid particles (sugar, cocoa powder, milk whey, milk powder, dehydrated cream, nut solids, etc.) in a continuous liquid (oil) or semi-solid phase (cocoa butter or other fats) [14,15]. Refining, also called grinding, aims to obtain an optimal size distribution of no fat solid particles and viscosity. For small-scale production, stirred ball mills represent a valid technology for cream production [16,17,18,19,20,21]. Ingredients and balls stirring in the mill tank result in impact and shearing actions that provide a progressive reduction of the no fat solid particles as well as their homogeneous dispersion [22]. The particle size distribution of creams depends on the amount and type of the solid particles and their size before grinding, but also on the oil/powder ratio, because if the oil/powder ratio is too low, the cream will be coarse [23]. The physical properties of refined creams are also affected by the physical state of the fat phase. Comparing the fineness of a hazelnut-and-cocoa-based paste, containing a vegetable oil, with that of a white chocolate flavouring paste, containing a blend of vegetable oil and fats, refined in the same way and containing powders of comparable initial granulometry, a similar evolution of fineness over refining time was observed [14,22]. However, those products differed in the type of oil/fat used, so presented a different PSD of creams at 25 °C, probably because in one of them there was also a high fraction of several fats that could crystallise when the temperature changed from 45 (processing in the ball mill) to 25 °C (storage).

For confectionary products, a few studies have reported the usage of oleogels in chocolate products, as summarised in the review of Zhao et al. (2021) [24]. Li and Liu (2019) [25] suggested the application of oleogels based on monoglyceril stearate, β-sitosterol/lecithin and ethyl cellulose to replace cocoa butter in dark chocolate. Results reported by Sun et al. (2021) [26] showed that oleogels with β-sitosterol and gamma oryzanol were the best substitutes for cocoa butter in dark chocolate. Recently, Bascuas et al. (2021) [27] designed new chocolate spreads with oleogels based on olive and sunflower oil, HPMC and xanthan gum. Their results showed that if the cocoa butter was replaced at 50% with the suggested oleogel, the final product was like the control one, but a complete replacement determined a quite different product, with a less homogeneous spread. To date, very few studies exist regarding using wax oleogels in chocolate products, and the oleogel has not been added at the beginning of refining ever. Doan et al. (2016) [28] used rice bran oil oleogel at different beeswax concentrations to partially replace the palm oil in hazelnut fillings. Fayaz et al. (2017) [29] studied the use of pomegranate oleogels–palm oil mixtures to produce functional chocolate spread. However, cream production involved the refining of only hazelnuts and sugar in a 3-roll mill, while the oleogel was later added by mixing with a mixer equipped with a “B” flat beater or by hand. 

Regarding the natural sweetener Lucuma, its chemical composition and beneficial effect on health have been largely studied [8,9], but only one study reported its inclusion in a confectionary product, a low-sugar chocolate [13]. Their results indicated that surface weighted mean D3,2 and mean particle size D50 should be less than 20 and 90 µm, respectively, to enhance chocolate textural quality. Moreover, the food application of lucuma, from a technological point of view, should be investigated. There are no studies that have investigated the effects of using oleogels, as well as Lucuma, on refining in a ball mill. Therefore, this work aimed to investigate the structure and physical properties of novel spreadable creams at different refining times. Firstly, the ability of the oleogel chosen to mimic cocoa butter’s behaviour during refining was verified. To do that, three different hazelnut and pumpkin seed oil creams were prepared in a stirred ball mill: the first one with cocoa butter, pumpkin seed oil and saccharose (CBS), the second one with oleogel [3] and saccharose (OS), and the third one containing oleogel and Lucuma powder as a partial saccharose replacer (OLS). The effect of formulation on the structure of creams at different refining times was investigated by analysing their granulometric and rheological behaviour. Oil binding capacity, water activity, colour and Turbiscan stability were also collected to assess how the physical stability of the creams, with different fat and/or sugar phases, can be modified during refining. The addition of both oleogel and Lucuma powder could allow the production of novel spreadable creams with improved health properties due to essential fatty acids and bioactive compounds.

## 2. Materials and Methods

### 2.1. Materials

The ingredients (hazelnut, cocoa, saccharose, cocoa butter and salt) used to prepare the creams were supplied by Me.Pa Alimentari S.r.l (Napoli, Italy). Pumpkin seed oil was purchased from Baule volante e Fior di Loto (Bologna, Italy). Micronized carnauba wax (CW) was kindly provided by a local candy company. Lucuma powder was purchased from Cibo Crudo Srl (Ciciliano, RM, Italy). Table 1 shows the nutritional values of Lucuma powder. The fatty acid composition of pumpkin seed oil was already reported by Borriello et al. (2022) [5].

#### Cream Preparation

The formulations of the three creams are shown in Table 2. For cream production, a Roboqbo universal processing system was used (model Qb8-4, Roboqbo s.r.l, Bologna, Italy) equipped with a spherical refining system (Bilia, designed to be used in conjunction with the Qbo—Universal Processing System series) with 4 kg of 6 mm diameter stainless steel AISI 316L balls. Ingredient weight was set at 3 kg, according to the instrument specifications. The speed rate was fixed at 250 rpm and the temperature was controlled at 45 ± 2 °C. Cutting of hazelnuts and sugar phases was performed at 25 °C for 15 min to obtain the hazelnut paste. Then, the other solid components and melted cocoa butter were added and mixed into the Qbo before starting the refining. An oleogel based on pumpkin seed oil and 6% carnauba wax was separately prepared, as reported by Borriello et al. (2021) [3]. The oleogelator was added to the oil previously heated at 90 °C and the mixture was stirred at 200 rpm, using a magnetic stirrer, until a clear solution was obtained. Then, the mixture was cooled down to 45 °C and finally added into the Qbo. Three productions for each cream were carried out, and samples refined for 60, 90, 120 and 150 min were collected. All the measurements were performed after 5 weeks of storage at 25 °C, except for the oil binding capacity measurements, which were performed one week after production.

### 2.2. Granulometric Measurement

The particle size distribution (PSD) of both saccharose and Lucuma powders was assessed by using a Mastersizer laser diffraction particle size analyser equipped with an Aero S dry powder dispersion unit (Malvern Instruments, Worcestershire, UK). The refining was stopped at 150 min since the solid particle size was close to 30 µm for all samples, measured by a digital micrometre (Metrocontrol Srl, Casoria, NA, Italy). The PSD of refined creams was analysed by using a Hydro 3000 wet sample dispersion unit. About 0.3 g of cream was analysed at ambient temperature (20 ± 2 °C) using sunflower oil as a dispersant to measure no fat solid particle size. For each cream, three different replicates were analysed and for each replicate, 20 measurements were performed.

### 2.3. Rheological Measurement

The rheological behaviour of refined creams was determined by a Modular Advanced Rheometer System (Haake MARS, Thermo Scientific, Waltham, MA, USA), equipped with a vane tool geometry (diameter 22 mm, length 16 mm, gap = 1 mm, ≈ 28 mL of the sample). Strain sweep tests (strain ranging from 0.0001 to 10%, frequency = 1 Hz) were carried out to define the LVR and to determine the yield stress. Frequency sweep tests (frequency ranging from 0.1 to 10 Hz, γ= 0.0005%) were performed to investigate the time-dependent deformation behaviour of the creams. The viscosity curves were measured as a function of the increasing shear rate (0.1–100 s^−1^) at 25 °C. The measurements were conducted in triplicate.

### 2.4. Oil Binding Capacity, Water Activity and Colour

The oil binding capacity was determined using a centrifuge method. Each cream (50 mL) was weighed in a centrifuge tube and centrifuged at 10,000 rpm for 20 min at 25 °C using a centrifuge (HERMLE Z 326 K). The excess oil was then decanted, and the weight of the remaining cream was determined. Oil binding capacity was calculated as:(1)$$\mathrm{OBC}\left(\%\right)=\left[1-\frac{\left(m_{1}-m_{2}\right)}{m_{1}}\right]\times 100.\;$$
where *m*_1_ is the mass of the initial sample and *m*_2_ is the mass of the cream after centrifugation. Three replicates were performed for each sample. Water activity determinations (Aw) were acquired using an Aqualab-Dew point water activity meter (4 TE, Decagon Devices Inc., Pullman, WA, USA). The colour of the creams was determined using a colourimeter (Minolta Chroma Meter, CR 300, Minolta, Tokyo, Japan). The Hunter parameters L* (from 0 = black to 100 = white), a* (−a = greenness to +a = redness), and b* (−b = blueness to +b = yellowness) were measured and averaged from three randomly positions.

### 2.5. Turbiscan Stability

The physical stability of creams was evaluated by measuring the backscattering (BS) of pulsed near infrared light (wavelength of 880 nm) using a Turbiscan Tower stability analyser (Formulaction, Toulouse, France). The cream was placed into cylindrical glass tubes up to the height of 45 mm and scanned for 120 h (≈5 days) at 25 °C. The stability of samples was expressed using the Turbiscan Stability Index (TSI), which is defined as follows:(2)$$TSI=\frac{\sum_{i}\sum_{h}\left|scan_{i\;}\left(h\right)-scan_{i-1\;}\left(h\right)\right|}{N_{h}}.\;$$
where *scan_i_* (h) is the light intensity of the scan acquired at the *i*-th time instant and a height of h, and Nh is the number of height positions in the selected scanning zone of the tube (top, centre, bottom or global) [22]. The data were analysed by using the software package TowerSoft Ver 1.2 (Formulaction, Toulouse, France).

### 2.6. Data Analysis

Results are reported as the mean ± standard error of at least three replications. Critical strain values (γ_0_) and yield stress (σ*) were determined from strain sweep curves. The critical strain was determined as the strain where *G*′ value decreased by more than 10% of the values recorded in the LVR, while the yield stress was defined as the stress where the viscous and elastic contributions were equal (*G*′ = *G*″) [30].

The frequency curves were fitted according to the rheological power-law model which is expressed by the following Equation (3) [31]:(3)$$G^{\prime }=a{\left(\omega \right)}^{b}.\;$$
where *G*′ is the storage modulus (Pa), *ω* is the frequency (rad/s), and *a* (Pa·sb) and *b* (−) are parameters used to describe rheological behaviour.

Apparent viscosity curves of spreadable creams were described using the Casson rheological model, which is adopted by the International Office of Cocoa and Chocolate for interpreting chocolate-based product behaviour, represented in the following Equation (4):

(4)$${\left(\eta \right)}^{1/2}={\left(\frac{\sigma_{0}}{\overset{\cdot }{ɣ}}\right)}^{1/2}+{\left(\eta_{\infty }\right)}^{1/2}$$
where *σ*_0_ is the Casson yield stress and η_∞_ is the infinite-shear viscosity, also called the Casson plastic viscosity. The yield stress can be used to calculate whether a sample is likely to settle in situ or whether it will be difficult to start pumping or stirring [31]. Multivariate analysis of variance (ANOVA) and multiple comparisons of means (Duncan’s test, *p* ≤ 0.05) were performed to evaluate whether differences among the samples, due to the different formulations (at the same refining time) and the different refining times (for each sample), were statistically significant (*p* ≤ 0.05) by using SPSS for Windows version 17.0 (SPSS Inc., Chicago, IL, USA).

## 3. Results

### 3.1. Particle Size Distribution

Sugar phases presented a different PSD (Appendix A); D90 values were 1190 µm and 190 µm for saccharose and Lucuma powder, respectively. However, the refining phase started after hazelnut paste was obtained (fineness of hazelnut paste ≈ 350 µm with or without Lucuma powder). Figure 1 shows the PSD of spreadable creams refined for 60 (Figure 1a), 90 (Figure 1b), 120 (Figure 1c), and 150 (Figure 1d) min.

As refining proceeded, the range of size classes of solid particles varied from 0.52–859 to 0.52–211 μm for cream CBS, from 0.52–515 to 0.52–211 μm for cream OS, and from 0.52–586 to 0.52–211 μm for OLS cream. The CBS and OS creams showed remarkably similar unimodal distribution, as also reported in other studies for anhydrous pastes refined in a ball mill refiner [14,22,29], with some differences at high particle size which became less evident as refining proceeded. OLS cream showed a larger particle size distribution curve with a lower particles volume percentage of around 11 μm compared to the other samples. Moreover, the PSD of the OLS cream changed from bimodal to unimodal during refining, due to the presence of a multicomponent sugar phase. At particle sizes between 0.5–0.8 μm, a small left shoulder was observed for all the samples. Appendix A reports the percentiles of the PSD for creams analysed. D10 did not discriminate among creams and was not affected by refining time. D50 and D90 values decreased as refining proceeded for all the creams. Figure 2 reported the evolution of D90 as a function of refining time, where 350 µm corresponded to the fineness of the coarsest fraction of unrefined cream, that is hazelnut paste.

Fineness and D90 tended to assume the same value for coarse particles [16,22]. CBS and OS creams refined for 150 min showed similar D90 values (26.4 μm and 24.4 μm) lower than those observed for sample OLS (37.4 μm). Those results seemed to suggest an effect of both oil and sugar phase on no fat solid particle size reduction after 60 min of refining, suggesting that oleogels seemed to accelerate the second part of the refining process; meanwhile, Lucuma powder had the opposite effect. All the creams presented the same oil/powder ratio, but the oil/sugar ratio in samples CBS and OS was lower than in sample OLS. Our results highlighted the effect of oil/sugar ratio on D90, as already observed by Armini et al. (2018) [23], for RUTF formulation. Even if the three unrefined creams presented the same fineness, we can hypothesise that the particle size distribution was different among creams, with a smaller number of coarse particles in cream OLS that should be reduced and a high number of fine particles that could slow down the refining process [32].

### 3.2. Rheological Properties

#### 3.2.1. Dynamic Rheological Measurements

Several sensory properties of spreadable creams, such as texture, firmness, smoothness, mouthfeel, and spreadability, depend on the fat crystal network [29]. Fat phase composition could affect the mechanical strength of the network, also influencing sensory attributes. To have information on the mechanical strength of the formed network, a rheological characterisation of the creams could be useful, by including both dynamic tests and viscosity determinations [33].

Strain sweeps (Figure 3a–d) were performed to determine the structural stability of spreadable creams by means of critical strain values (γ_0_) and yield stress (σ*) determinations, as shown in Table 3. Critical strain is the onset of nonlinearity corresponding to the structure deformation necessary to initiate flowing [34]. The yield stress, related to material stability, but also to hardness and spreadability, represents the point at which the deforming creams begin to show a liquid-like behaviour [35]. All the samples showed a gel-like behaviour with a storage modulus (*G*′) greater than the loss modulus (*G*″) and a linear viscoelastic region (LVR) approximately in the range of 0.0001–0.006%, in accordance with results reported by Palla et al. (2021) [36] for filling creams made with monoglyceride oleogels. There were no significant differences in critical strain (γ_0_) values observed for creams refined for the same amount of time, except a slightly lower γ_0_ value than the others in cream OLS at the end of refining. Independently of the refining time, different values of yield stress (σ*) were observed following this order: CBS < OLS < OS. However, refining time strictly affected both γ_0_ and σ* parameters (*p* > 0.05), which increased with increasing refining time. Those results seemed to suggest that creams with oleogels were more structured and required higher shear stress to flow, compared to the cream with cocoa butter. The yield stress is related to the strength of the network structure, and depends on attractive particle–particle interactions. Therefore, particle size and particle concentration, as well as the density of the network, could affect its magnitude [37,38]. Comparing creams with a similar D90 around 25 μm, e.g., CBS cream with OS cream refined for 150 min (Table 2), their σ* values (Table 3) were quite different. However, the solid particles, in our case, saccharose and cocoa powders, could be dispersed differently in the matrix depending on the fat matrix composition [36,39], so particle–particle interactions could vary. It could be hypothesised that in OS cream the interactions among solid particles inside carnauba wax and pumpkin seed oil, lead to a strong three-dimensional network. On the other hand, cocoa butter decreases forces between particles, reducing the resistance to flow and leading to lower values of yield stress [40]. Lucuma powder seemed to hinder the affinity between solid particles, probably due to its heterogeneous composition.

At the lowest refining time, creams OS and OLS showed similar *G′* and *G″* close to 40,000 Pa, values which were higher than those observed for sample CBS (10,000 Pa). As the refining proceeded, *G*′ values increased with a linear trend (R^2^ = 0.99; m = 1722.4) in sample CBS, with an exponential trend (R^2^ = 0.99; n = 0.019) in sample OS. Finally, *G′* value of sample OLS linearly increased (R^2^ = 0.99; m = 611.62) up to 120 min of refining, then reached a constant value. Sample OLS refined for 150 min showed a slightly lower storage modulus than the other samples, mainly due to the low saccharose amount. A direct relationship between the *G′* and sugar amount was also found by Palla et al. (2021) [36] in filling creams containing monoglyceride oleogel as fat phase. Those results, in accordance with PSD profiles, suggested that for the OS cream the refining process should be stopped at 120 min, since desired values of both particle size and yield stress were obtained.

The mechanical spectra of the frequency sweep of creams were acquired and fitted with the power-law model, estimating parameters *a* and *b* (Table 3). The power-law model (Equation (3)) describes storage modulus data (R^2^ > 0.92). For all refining times, cream CBS showed the lowest *a* value, which ranged from 15547 to 36448 Pa·s ^b^, while *b* values were like those estimated for sample OLS (≈0.19). Cream OS always showed the highest *a* value and the lowest *b* value compared to the other samples. Moreover, a relationship between refining time and both parameters was found in CBS cream (*a p* = 0.00; *b p* = 0.03), OS cream (*a p* = 0.00; *b p* = 0.001) and OLS (*p* = 0.00 for *a* and *b*). In detail, *a* increased with refining time, while *b* followed the reverse order. The solid-like behaviour was confirmed for all the samples, which showed a slight frequency dependence and a good tolerance to deformation rate. Fayaz et al. (2017) [29] also found a solid-like behaviour in chocolate spreads prepared with palm oil and wax oleogels in the ratio of 1:1. After 60 min of refining, the OS and OLS creams exhibited similar storage moduli, which were higher than those observed in the CBS sample. Similar values of storage and loss moduli were also found by Glicerina et al. (2013) [33] in nut creams with distinct types of fats. In detail, samples of OS and OLS could resemble a cream with 1:1 and 3:1 palm oil/hydrogenated fat ratios, respectively.

#### 3.2.2. Steady Shear Measurement

Knowledge of the apparent viscosity of a cream is crucial both in quality control of the product and in process application. The apparent viscosities of spreadable creams refined for various amounts of time are represented in Figure 4. All the spreads showed pseudo-plastic behaviour since the apparent viscosity (η) decreased as the shear rate ($\overset{\cdot }{ɣ}$) increased. The viscosity of CBS and OS creams increased as the refining time increased [17], meanwhile, for OLS cream, this behaviour was observed only moving from 60 (Figure 4a) to 90 min (Figure 4b) of refining time. After 120 min (Figure 4c), CBS and OLS creams showed remarkably similar viscosity curves, lower than that of OS cream. However, after 150 min (Figure 4d), OLS cream seemed to have the lowest viscosity values, indicating an effect of the Lucuma powder on this cream property, as also observed by Miele et al. (2020) [41] for RUTF formulations in which the oil/powder ratio was the same, but the sugar content was reduced, and the soy flour content increased. However, it is also true that at the end of the refining process, OLS cream presented a D90 higher than that of both CBS and OS creams.

The Casson model (Equation (4)) well fitted flow curves (R^2^ = 0.998), the estimated parameters for Casson yield stress (σ_0_) and the Casson plastic viscosity (η_∞_) are reported in Table 4. At the lowest refining time, all the samples showed similar η_∞_ values close to 8 Pa·s. After 120 min of refining, CBS and OS creams showed the highest (≈14 Pa·s) and the lowest (≈5 Pa·s) plastic viscosity, respectively. A higher plastic viscosity can cause problems during the process of some types of chocolate formulations [42]. In the CBS and OLS samples, the Casson plastic viscosity increased as refining proceeded from 7 to 14 and from 7 to 10 Pa·s, respectively. However, in OS cream, η_∞_ seemed to decrease with particle size reduction. Afoakwa et al. (2008) [43] reported an increase in plastic viscosity and decreasing particle size for chocolate samples with 25% fat and 0.3% lecithin, due to the increased points of contact between particles. However, they also noted a reduction in plastic viscosity with increasing lecithin from 0.3 to 0.5%, especially at lower particle sizes. They attributed this reduction to an association between lecithin and sugar particles. As already stated, the fat phase could affect sugar dispersion with an effect on the viscosity and rheological properties of the final product [36,39], because when sugar crystals were well dispersed in the matrix, the viscosity of the product was reduced. In our case, the oleogels, or more specifically the carnauba wax, seemed to behave as lecithin, reducing plastic viscosity at low particle size.

A significant increase in Casson yield values at decreasing particle sizes was found, according to many investigators [43,44]. Sample OS showed higher σ_0_ values than sample CBS at all refining times. A similar result was also found by Patel et al. (2014) [42] in chocolate spreads with shellac oleogels. They found that spreads with oleogel had higher yield stress than reference chocolate paste prepared using an oil binder due to the higher interactions between solid sugar particles. Furthermore, Taylor et al. (2009) [45] investigated the rheological behaviour of chocolate with crumb and sunflower oil, finding σ_0_ and η_∞_ values in the range of 20–70 Pa and 4–15 Pa·s, respectively. Our results are also in line with those reported by Cavella et al. (2020) [22] who studied the effect of ball milling on the rheological properties of a white chocolate anhydrous paste with a fat phase made up of a mix of cocoa butter, palm, and sunflower oil, and Loncarevič et al. (2017) [19], who suggested using sesame and rapeseed oil instead of sunflower oil in spreadable creams.

### 3.3. Oil Binding Capacity, Water Activity and Colour

The oil binding capacity plays a key role in determining the effectiveness of a new fat formulation [28]. Oil binding capacity (OBC), water activity (Aw), and colour parameters (L* a* b*) are reported in Table 5. Sample OS and OLS showed similar OBC values (≈86%), which were slightly lower than those observed for CBS cream (≈89%). Doan et al. (2016) [28] reported that most of the oil loss occurred within the first day after production and then decreased. Considering that OBC measurements were performed one week after sample production, and the total replacement of cocoa butter led to an oil loss of only 4% greater than that observed for cream CBS, it can be concluded that samples containing OPCW oleogel exhibited a particularly good ability to retain oil. Moreover, the refining did not affect the OBC of samples, although for creams refined for 120 min, slightly higher OBC values were observed. Samples were microbiologically stable since their aw values were close to 0.5. There were no significant differences between the L*, a* and b* values of samples CBS and OS, while OLS cream showed lower L* and b* values compared to the others. The lightness increased with the refining time. On the other hand, Lucuma powder decreased both the lightness and yellowness of the cream.

### 3.4. Turbiscan Stability Index

The physical stability of spreadable creams could be affected by changes in the fat or sugar phase. Figure 5a–d report the evolution over time of the Turbiscan stability index (TSI) for the three creams at different refining times. Samples of CBS and OS initially showed a similar TSI trend that rapidly increased in first 7 h, then the growth rate slowed down. The OLS cream followed an opposite trend since it initially showed the lowest TSI kinetics, which became faster to reach TSI values like OS cream after 5 days of storage. At the lowest refining degree, the TSI curve of the OLS sample showed a characteristic shape, indicating that the sedimentation had occurred in two steps because of the larger particle size population in different sweeteners. Accordingly, with PSD curves, bimodal behaviour faded as refining proceeded. The TSI value of sample CBS and OLS decreased from 1.32 to 0.89 and from 0.9–0.75 with increasing refining time, respectively. The cream with cocoa butter as the lipid phase showed the highest TSI values, mainly due to the sedimentation of solid particles but still acceptable as lower than 1.4. There were no differences between the TSI value (0.7) of the OS sample refined for 90, 120 and 150 min of refining. Moreover, the OS sample was the only one that assumed a constant trend, suggesting that it will not be subject to significant destabilisation phenomena over long storage time. However, all the samples were stable during about 5 days of storage since TSI values were lower than 10, which is considered the Turbiscan stability threshold. Our results are in line with those reported by Cavella et al. (2020) [22], which investigated the TSI at the top of the vials of anhydrous pastes with cocoa butter refined in stirred ball mills. They showed that the samples refined for 150 min reached a TSI value close to 4 in about 70 h and that the sedimentation occurred mainly for the less refined sample. All of the creams analysed in this work showed a TSI value lower than 1.4 and a TSI at the top of the vials (data not shown) lower than 1.2, suggesting that both the particle sedimentation and the consequent separation of oil phase at the meniscus were negligible. Babin et al. (2005) [39] report that sugar crystals interact differently with each other depending on the fat phase, determining different sediment volumes, which could explain the slight differences observed in the TSI values of our samples. They reported that a lower sediment volume was observed in palm kern oil than in cocoa butter because crystals were more aggregated in the first fat, while they interacted with repulsive forces in the cocoa butter.

## 4. Conclusions

The present work emphasised the impact of a novel oleogel and an uncommon sweetener on the structure and physical properties of novel spreadable creams at different refined degrees. Similar creams were obtained by changing only the fat phase in terms of PSD and rheological properties, with slight differences in terms of the strength of the network structure. The oleogel accelerated the second part of the refining process and its structure was completely recovered in cream with 100% saccharose, which showed high shear stress to flow. At low particle size, the carnauba wax acted as an emulsifier, reducing the Casson plastic viscosity. On the other hand, Lucuma seemed to slow down the refining and slightly hinder the recovery of the oleogel structure in the cream. A solid like-behaviour and a good tolerance to deformation rate were observed for all the samples, showing that the oleogel based on pumpkin seed oil and carnauba wax could totally replace the cocoa butter, and Lucuma could partially decrease the saccharose intake. Moreover, both oleogel and Lucuma did not affect oil binding capacity and water activity, ensuring the physical and microbiological stability of creams. Our results confirmed the effect of particle size on rheological and stability properties also in oleogel-based creams. Concluding, these findings could provide useful information in the application of pumpkin seed oil oleogel and Lucuma powder for novel confectionery products with healthier nutritional profile.

## Figures and Tables

**Figure 1 foods-11-01844-f001:**
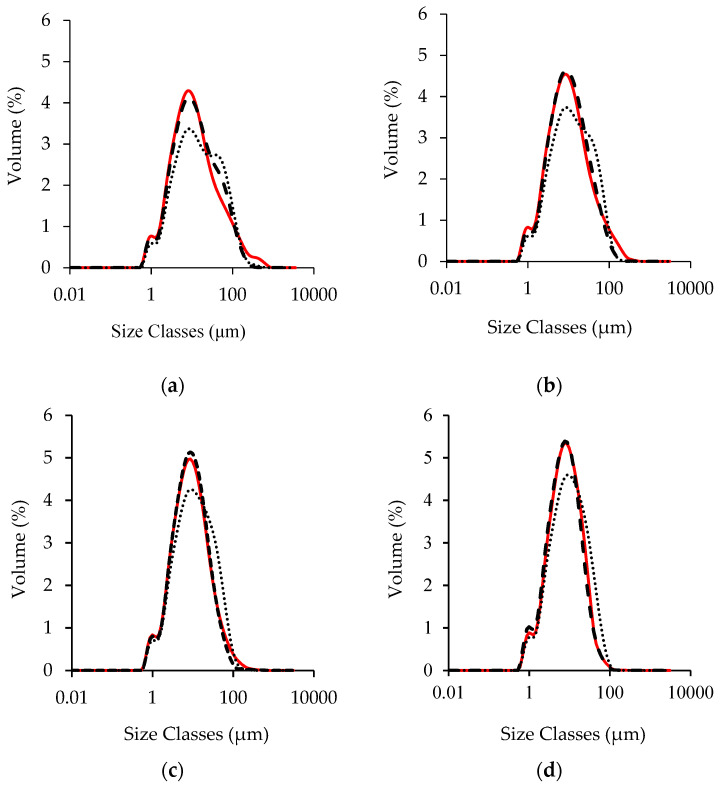
Particle size distribution of spreadable creams CBS (

), OS (
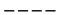
) and OLS (
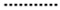
), at different refining times 60 (**a**), 90 (**b**), 120 (**c**) and 150 (**d**) min.

**Figure 2 foods-11-01844-f002:**
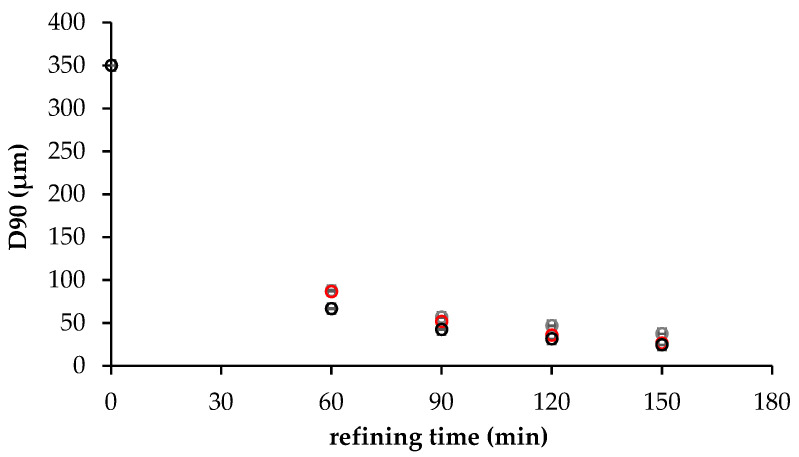
D90 of spreadable creams CBS (●), OS (■), OLS (
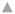
) as a function of refining times.

**Figure 3 foods-11-01844-f003:**
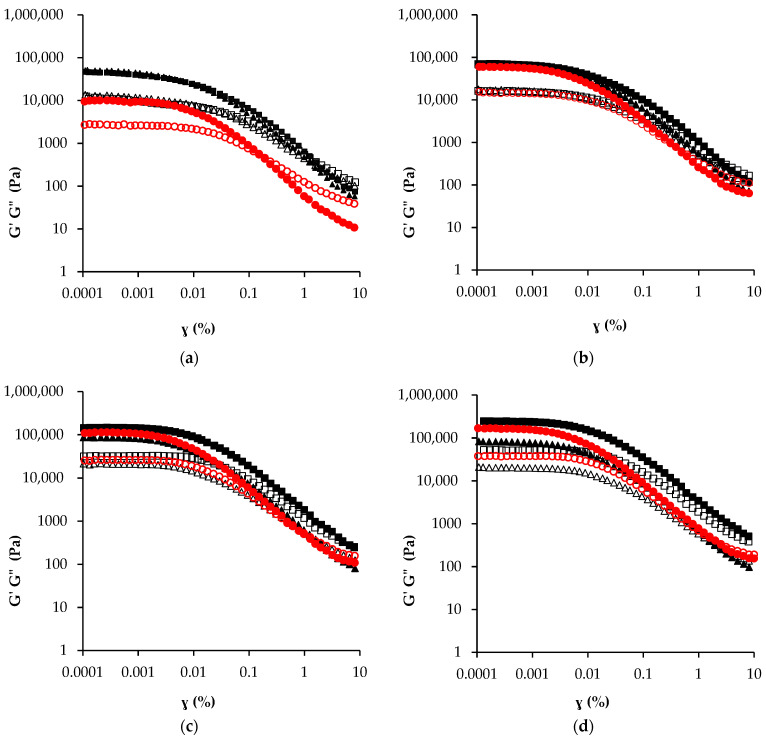
Strain sweeps of spreadable creams CBS (●), OS (■), OLS (▲) at different refining times 60 (**a**), 90 (**b**), 120 (**c**) and 150 (**d**) min. Fill indicators *G*′, empty *G*″.

**Figure 4 foods-11-01844-f004:**
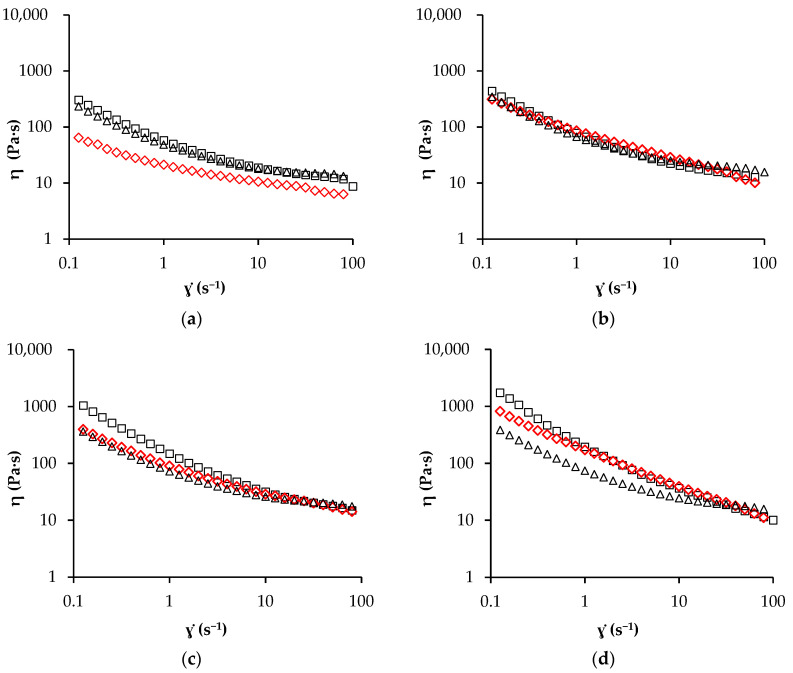
Apparent viscosity (η) as a function of shear rate ($\overset{\cdot }{ɣ}$) for spreadable creams CBS (◇), OS (□) and OLS (Δ) refined for different refining times 60 (a), 90 (b), 120 (c) and 150 (d) min.

**Figure 5 foods-11-01844-f005:**
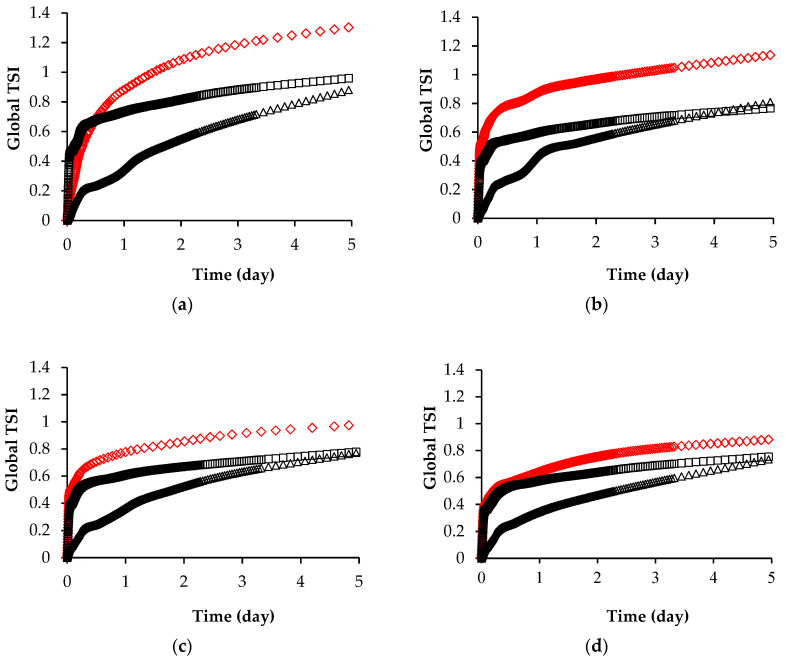
Turbiscan stability index (TSI) as function of storage time (day) for spreadable creams CBS (◇), OS (□) and OLS (Δ) refined for 60 (**a**), 90 (**b**), 120 (**c**) and 150 (**d**) min.

**Table 1 foods-11-01844-t001:** Nutritional values of Lucuma powder.

Lucuma Powder	
Protein	5 g
Lipids	4 g
Carbohydrates	25 g
Fibers	3.6 g
Calcium	92 mg
Phosphorus	186 mg
Iron	4.6 mg
Vitamin C	11.6 mg
Riboflavin	0.3 mg
Thiamine	0.2 mg

**Table 2 foods-11-01844-t002:** Spreadable creams composition (%).

Ingredients	Formulations		
	CBS	OS	OLS
Hazelnut	45	45	45
Cocoa powder	4.9	4.9	4.9
Saccharose	32	32	16
Lucuma powder	-	-	16
Cocoa butter	10	-	-
Pumpkin seed oil	8	8	8
Oleogel OPCW6	-	10	10
Salt	0.1	0.1	0.1

**Table 3 foods-11-01844-t003:** The strain at the limit of linearity (γ_0_) and the yield stress (σ*) determined by strain sweep test; the parameters a and b estimated fitting frequency curves with the power-law model (G′ = a ω^b^).

Sample	Refining Time (min)	γ_0_ (%)	σ* (Pa)	*a* (Pa·s ^b^)	*b*	R^2^
CBS	60	0.002 ± 0.00 ^a^	1.16 ± 0.00 ^a^	15,547 ± 1332 ^a^	0.197 ± 0.00 ^b^	0.98
	90	0.001 ± 0.00 ^a^	4.29 ± 0.27 ^b^	20,241 ± 1004 ^b^	0.192 ± 0.00 ^b^	0.97
	120	0.001 ± 0.00 ^a^	6.76 ± 0.30 ^c^	22,236 ± 968 ^b^	0.188 ± 0.00 ^b^	0.98
	150	0.006 ± 0.00 ^b^	11.76 ± 0.01 ^d^	36,448 ± 1452 ^c^	0.115 ± 0.01 ^a^	0.96
OS	60	0.002 ± 0.00 ^a^	8.12 ± 0.50 ^a^	39,616 ± 4146 ^a^	0.175 ± 0.05 ^c^	0.98
	90	0.002 ± 0.00 ^a^	12.33 ± 0.48 ^b^	62,057 ± 4257 ^b^	0.152 ± 0.05 ^b^	0.97
	120	0.003 ± 0.00 ^b^	23.11 ± 0.14 ^c^	141,118 ± 2375 ^c^	0.149 ± 0.02 ^b^	0.92
	150	0.005 ± 0.00 ^c^	51.33 ± 0.05 ^d^	219,261 ± 5425 ^d^	0.131 ± 0.01 ^a^	0.97
OLS	60	0.001 ± 0.00 ^a^	6.46 ± 0.09 ^a^	38,751 ± 753 ^a^	0.204 ± 0.02 ^d^	0.98
	90	0.002 ± 0.00 ^b^	7.31 ± 0.00 ^b^	53,519 ± 875 ^b^	0.193 ± 0.01 ^c^	0.97
	120	0.001 ± 0.00 ^a^	8.19 ± 0.00 ^c^	72,452 ± 1008 ^c^	0.177 ± 0.02 ^b^	0.97
	150	0.002 ± 0.00 ^b^	9.18 ± 0.00 ^d^	73,870 ± 2265 ^c^	0.169 ± 0.00 ^a^	0.97

Different letters correspond to significantly different values (*p* < 0.05) from Duncan’s test. The letters are missing where no significant differences between the samples were observed.

**Table 4 foods-11-01844-t004:** Rheological parameters (mean ± standard error) of Casson model (yield stress, σ_0_, and viscosity at infinite shear rate, η_∞_), for CBS, OS and OLS creams at different refining times.

Sample	Refining Time (min)	Casson Parameters		
		σ_0_ (Pa)	η_∞_ (Pa·s)	R^2^
CBS	60	4.15 ± 0.01 ^a^	7.85 ± 0.33 ^a^	0.99
	90	25.45 ± 2.33 ^b^	13.04 ± 2.32 ^ab^	0.99
	120	37.00 ± 0.00 ^c^	10.23 ± 0.00 ^bc^	0.99
	150	76.82 ± 0.07 ^d^	14.53 ± 0.86 ^c^	0.99
OS	60	26.20 ± 3.05 ^a^	8.36 ± 0.30 ^b^	0.99
	90	37.20 ± 0.00 ^b^	7.86 ± 0.02 ^b^	0.99
	120	83.10 ± 0.10 ^c^	7.72 ± 0.14 ^b^	0.99
	150	158.66 ± 0.03 ^d^	4.91 ± 0.82 ^a^	0.98
OLS	60	15.40 ± 3.74 ^a^	7.72 ± 0.15 ^a^	0.99
	90	29.77 ± 0.00 ^b^	8.86 ± 0.00 ^b^	0.99
	120	29.95 ± 0.00 ^b^	9.51 ± 0.00 ^b^	0.99
	150	32.11 ± 0.00 ^b^	10.36 ± 0.43 ^c^	0.99

Different letters correspond to significantly different values (*p* < 0.05) from Duncan’s test. The letters are missing where no significant differences between the samples were observed.

**Table 5 foods-11-01844-t005:** Spreadable creams’ physical characterization. Oil binding capacity (OBC %), water activity (%) and colour parameters (L* a* b*) of creams CBS, OS and OLS at different refining times.

Sample	Refining Time (min)	OBC	Aw	L*	a*	b*
CBS	60	86.66 ± 0.77 ^a^	0.50 ± 0.00	23.05 ± 0.11 ^a^	14.56 ± 0.10	10.53 ± 0.10 ^a^
	90	90.44 ± 0.22 ^c^	0.49 ± 0.00	24.17 ± 0.05 ^b^	14.81 ± 0.03	11.36 ± 0.09 ^b^
	120	90.66 ± 0.00 ^c^	0.50 ± 0.01	25.18 ± 0.08 ^c^	14.71 ± 0.05	12.36 ± 0.12 ^c^
	150	88.66 ± 0.38 ^b^	0.48 ± 0.01	27.01 ± 0.04 ^d^	14.38 ± 0.10	13.37 ± 0.08 ^d^
OS	60	85.77 ± 1.17	0.49 ± 0.02	23.24 ± 0.10 ^a^	14.48 ± 0.14	10.31 ± 0.22 ^a^
	90	85.77 ± 0.44	0.51 ± 0.00	24.14 ± 0.06 ^b^	14.51 ± 0.07	11.01 ± 0.08 ^b^
	120	86.22 ± 1.16	0.45 ± 0.01	26.09 ± 0.42 ^c^	13.95 ± 0.22	11.50 ± 0.34 ^b^
	150	85.77 ± 1.09	0.48 ± 0.01	26.89 ± 0.14 ^d^	14.10 ± 0.03	12.61 ± 0.03^c^
OLS	60	86.50 ± 0.95	0.44 ± 0.02	20.26 ± 0.85	14.23 ± 0.07	7.63 ± 0.93
	90	86.00 ± 1.15	0.42 ± 0.00	19.74 ± 0.02	14.31 ± 0.07	6.67 ± 0.11
	120	87.33 ± 0.38	0.43 ± 0.00	20.66 ± 0.00	14.38 ± 0.10	7.30 ± 0.05
	150	86.66 ± 0.77	0.43 ± 0.00	21.10 ± 0.03	14.25 ± 0.01	8.20 ± 0.11

Different letters correspond to significantly different values (*p* < 0.05) from Duncan’s test. The letters are missing where no significant differences between the samples were observed.

## Data Availability

Data is contained within the article or Appendix A.

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
