# Peer review of "Rheological Properties, Particle Size Distribution and Physical Stability of Novel Refined Pumpkin Seed Oil Creams with Oleogel and Lucuma Powder"

_foods, 2022, doi:10.3390/foods11131844_

Round 1
Reviewer 1 Report
Comments and Suggestions for Authors
The article „Rheological properties and physical stability of novel refined pumpkin seed oil creams with different fat and sugar phases“ is focused on the physico-chemical properties of spreadable creams prepared using a new recipe.
Although the presented products in this article have a potential application, there are some issues that must be improved:
Line 2: It seems that the one aim of this work was the use of oleogel and lucuma powder. However, the title does not reflect this fact.
Line 112: Did you also provide sensory analysis of prepared products? The sensory properties play a significant role in the assessment of these types of products …
Line 114: specify sugar – saccharose?
Line 121: Table 1: I suggest Vitamin C instead of the Vit.c, Riboflavina and Thiamina are not english words …
Line 123: Did you control a microbiological profile of creams?
Line 138: Why did not you prepare the formulation with lucuma powder only (completely without „sugar“)?
Line 140 – 173: When were these measurements (2.2. – 2.4.) realized – immediately after preparation? Especially for these creams, it would be useful to evaluate the properties during a longer storage period (e.g. 1 month) …
Line 249 – 251: I suppose the value 0.52 is a lower limit of the device …
Line 280: I suggest to add a discussion how the measured values of the rheological parameters influence the sensory profile of products and what are usual preferences of consumers…
Author Response
We greatly appreciate the reviewers’ constructive comments and suggestions that are very helpful for revising and improving our paper. We have checked and revised the manuscript carefully according to the comments. The revisions to the manuscript were marked up using the track change function. Please see the attachment.

Reviewer 2 Report
I suggest removing the word "phases" from the title. Instead use "composition" or "Ratio". Also, there is no mention of particle size in the title. As far as I can see, you stated that the aim was to also to analyze the impact of particle size on rheology. Please rephrase the title accordingly.
The paper should be proof read by an English professional. It contains grammatically incorrect sentences, as well as missing words in some sentences.
P3, Table 1. Why is only lucuma powder composition shown in this table? You use many different complex food materials, it makes no sense that you presented the composition of only one. Add the composition of other ingredients or remove table 1.
P7, Figure 2: Model approximation based on only 5 points is not valid. Namely, you did not collect samples in the time period from 0 to 60 mins in which the greatest changes in particle sizes happen, and have, therefore, missed the slope of the curve which is important for model approximation. This data is simply not valid for modeling and the conclusions drawn thereof are not sound.
P12, Table 5: statistical significance data is missing for some parameters (aw and a*).
P5, L204-208: Considering the low sample number, did you test the normality of data distribution to be sure you can perform ANOVA on your data?
Author Response

(The authors gave the same response as above.)

Round 2
Reviewer 1 Report
Thank you for your responses and additional explanations.